# The Transcriptional Responses of Ectomycorrhizal Fungus, *Cenococcum geophilum,* to Drought Stress

**DOI:** 10.3390/jof9010015

**Published:** 2022-12-21

**Authors:** Mingtao Li, Chao Yuan, Xiaohui Zhang, Wenbo Pang, Panpan Zhang, Rongzhang Xie, Chunlan Lian, Taoxiang Zhang

**Affiliations:** 1International Joint Laboratory of Forest Symbiology, College of Forestry, Fujian Agriculture and Forestry University, Fuzhou 350002, China; 2Forestry Bureau, Sanyuan District, Sanming 365000, China; 3Asian Research Center for Bioresource and Environmental Sciences, Graduate School of Agricultural and Life Sciences, The University of Tokyo, 1-1-1 Midori-cho, Nishitokyo, Tokyo 188-0002, Japan

**Keywords:** *Cenococcum geophilum*, drought, fungal transcriptomics, peroxisome

## Abstract

With global warming, drought has become one of the major environmental pressures that threaten the development of global agricultural and forestry production. *Cenococcum geophilum* (*C. geophilum*) is one of the most common ectomycorrhizal fungi in nature, which can form mycorrhiza with a large variety of host trees of more than 200 tree species from 40 genera of both angiosperms and gymnosperms. In this study, six *C. geophilum* strains with different drought tolerance were selected to analyze their molecular responses to drought stress with treatment of 10% polyethylene glycol. Our results showed that drought-sensitive strains absorbed Na and K ions to regulate osmotic pressure and up-regulated peroxisome pathway genes to promote the activity of antioxidant enzymes to alleviate drought stress. However, drought-tolerant strains responded to drought stress by up-regulating the functional genes involved in the ubiquinone and other terpenoid-quinone biosynthesis and sphingolipid metabolism pathways. The results provided a foundation for studying the mechanism of *C. geophilum* response to drought stress.

## 1. Introduction

Due to global warming, the increase in temperature reduces the annual water content of soil, making droughts in arid and semi-arid areas more serious [1]. Drought stress is the most common environmental factors limiting plant growth and ecosystem productivity [2]. Droughts have adverse effects on many biological activities in plants, such as photosynthesis, nutrient acquisition, and cell metabolism, resulting in decreased chlorophyll concentration, cell membrane stability, leaf water content, and increased oxidative damage [3,4,5]. Severe droughts can even cause widespread death of vegetation in forests [6]. It is reported that droughts have triggered the mortality of *Pinus tabulaeformia* across 0.5 million hectare in east-central China [6]. The reduction of tree production caused by drought stress may exceed the total reduction caused by other environmental stresses, resulting in incalculable social and economic losses [7]. At present, the mycorrhizal afforestation technology which can improve the resistance of forest ecosystem to abiotic stress has attracted more and more attention [8,9,10,11].

Mycorrhiza is a symbiosis formed by fungi and plant roots, and more than 80% of vascular plants are able to form mycorrhizal symbiosis with fungi [9]. Fungi provide water and mineral elements, such as nitrogen and phosphorus, to plants, while plants provide carbohydrates for fungi [10]. In recent years, more and more studies have shown that mycorrhizal fungi can improve the physiological performance of plants under drought conditions, such as enhancing water and nutrient absorption, promoting photosynthesis and transpiration, activating the anti-oxidative stress system of plants, and maintaining cells’ osmotic pressure [11,12,13,14]. In addition, mycorrhizal fungi are able to delay stomatal closure caused by drought [15], improve leaf stomatal conductance, and increase gas exchange, so as to help host plants adapt to drought environment [16]. Drought stress causes the imbalance of the production and clearance of reactive oxygen species (ROS) in plants, resulting in the cumulative damage of ROS to cell membranes [17,18]. It was found that the inoculation of mycorrhizal fungi reduced the accumulation of MDA and hydrogen peroxide in plants under drought stress, which enhanced the drought resistance of plants [12]. Moreover, mycorrhizal fungi also activate the activities of catalase (CAT), superoxide dismutase (SOD), peroxidase (POD), glutathione peroxidase (GPX), and ascorbate peroxidase (APX) to increase the adaptability of plants to drought stress [19].

Aquaporin is a multifunctional protein family, which widely exists in almost all organisms [20]. The existence of aquaporins is conducive to the transport of small molecules, including water, carbon dioxide, glycerol, amino acids, and even short peptides, ions, and metal-like substances, which may play a direct or indirect role in the water regulation of plants under drought stress [21,22]. At the same time, the aquaporin family can increase the osmoregulation ability of organisms, such as PIP and TIP in the aquaporin family participate in radial transcellular water transport and cell osmoregulation in higher plants [20]. A large number of studies have shown that mycorrhizal fungi can optimize the expression pattern of aquaporin genes in plants, so that mycorrhizal plants have a better water relationship than non-mycorrhizal plants under drought stress [23,24,25]. Fungal SOD is an antioxidant enzyme that scavenges ROS and also participates in defense responses in mycorrhizal plants [26]. A previous study has found that a GintSOD1-encoding Cu/Zn-SOD maintains the function of scavenging ROS in mode fungus *Glomus intraradices* [27]. Benabdellah et al. found that the expression level of GintPDX1 in fungi increased after exogenous ROS induction, suggesting that GintPDX1 plays a role in protecting fungi from oxidative damage and anti-oxidation [28,29]. Moreover, most studies have confirmed that the inoculation of mycorrhizal fungi can up-regulate the relative expression levels of PtFe-SOD, PtMn-SOD, PtPOD, and PtCAT1 genes in plants in response to drought stress [14,30].

*Cenococcum geophilum Fr. (C. geophilum)*, belonging to Ascomycota, is a common ectomycorrhizal fungus in forest systems, which can form ectomycorrhizal roots with a large variety of hosts, such as Betulaceae, Fagaceae, and Pinaceae [31]. Because *C. geophilum* is highly tolerant to abiotic stresses, such as drought, salinity, heavy metals, and high temperatures, this species dominantly grows in sand dunes, moraines, volcanic ash, cinder, and other harsh habitats [32,33]. In addition, this fungus can help host plants absorb nutrients and water, promote the growth of plants, and resist environmental stress, such as drought and salinity [34]. It was reported that *C. geophilum* grew well at NaCl concentrations of up to 500 mM and 2 mg/L of cadmium chloride in a MMN medium [35,36]. Jany et al. found that *C. geophilum* is able to maintain the water conditions of Beech rhizosphere soil facing drought stress [37]. Under salt stress, the inoculation of *C. geophilum* could increase the biomass of *Pinus thunbergii* seedlings and the nutrient elements in shoots [38,39]. Therefore, *C. geophilum* can be used as a reliable inoculation source for cultivating drought-tolerant mycorrhizal seedlings. However, the mechanism of drought resistance in *C. geophilum* and the genes involved in drought resistance are still unclear.

In this study, six strains (three drought-tolerant; three drought-sensitive) of *C. geophilum* were selected to analyze the molecular responses to drought stress with treatment of 10% poly-ethylene glycol (PEG-6000). After drought stress, physiological indicators and RNA sequencing were performed to identify key differentially expressed genes in *C. geophilum* for drought stress resistance. The results would provide a foundation for studying the molecular mechanism of *C. geophilum* response to drought stress.

## 2. Materials and Methods

### 2.1. Mycelial Growth of Different Strains of C. geophilum under Drought Treatments

Three drought-sensitive strains (Chcg01, Jacg40 and Jacg45) and three drought-tolerant strains (Jacg37, Jacg153 and Jacg205) evaluated in previous experiments were used in this study (Appendix A). The six strains were pre-cultured in a modified Melin–Norkrans (MMN) agar medium [40] at 25 °C in the dark for 45 days. The pure culture of *C. geophilum* is the same as described in the previous reports by Shi et al. [35] and Li et al. [36]. During the drought treatment, 10% polyethylene glycol (PEG-6000) was used to simulate drought stress. An agar plug with *C. geophilum* (7 mm in diameter) cultured for two months was transferred into a 90 mm petri dish containing 10% PEG-6000 liquid medium (10 mL) covered with a cellophane membrane, and cultured for 30 days in the dark at 25 °C. Each strain had three replicates for all treatments. After culturing for 30 days, the mycelial area of each strain was measured using a planimeter (X-Plan 380d III; Kantum Ushikata, Yokohama, Japan).

### 2.2. Preparation of Mycelia of C. geophilum for the Determination of Physiological Indicators

The mycelia of the six *C. geophilum* strains were precultured in a liquid MMN medium (50 mL). After 30 days of dark incubation, 12.5 mL of 50% PEG-6000 was added to the liquid MMN medium, with a final concentration of 10% PEG-6000, and then the mycelia were cultured for further two weeks. At the same time, 12.5 mL of PEG-6000-free liquid MMN medium was added to each flask of the control group (CK). Three biological replicates were conducted for each strain. The mycelia were collected from each conical flask by filtration and washed with 0.01 M PBS. The procedures of mycelium culture were described previously by Shi et al. [35] and Li et al. [36]. Then, the collected mycelium was divided into two parts. One part was rapidly frozen in liquid nitrogen and stored in a −80 °C refrigerator to determine the activities of POD (peroxidase), SOD (superoxide dismutase), and CAT (catalase). The remaining part was used for the determination of nutrient content after drying at 105 °C for 24 h.

### 2.3. The Determination of Antioxidant Enzyme Activity in C. geophilum Mycelia

The activities of superoxide dismutase (SOD), peroxidase (POD), and catalase (CAT) were determined by the kits provided by the Nanjing Jiancheng Institute of Biological Engineering, and three biological replicates were conducted for each indicator.

### 2.4. The Determination of Na, Ca, P, and K Concentrations in C. geophilum Mycelia

The dried mycelia of *C. geophilum* (about 0.1 g) were digested with nitric acid (HNO_3_) and hydrogen peroxide (H_2_O_2_) (v:v = 5:1) in an automatic microwave digestion system for 1.5 h. The digestion solution was added at constant volume to 25 mL with ultra-pure water, and then filtered with a 0.45 micromole filtration membrane. The concentrations of Na, K, Ca, and P in the digestion solution were determined using an inductively coupled plasma-optical emission spectrometer (ICP-OES, peopti-ma8000, PerkinElmer).

### 2.5. Preparation of Mycelia of C. geophilum Strains for RNA-Seq and Quantitative Real-Time PCR (qRT-PCR) Analyses

The six strains were cultured in a MMN agar medium covered with cellophane for 30 days, and then the mycelium was scraped off the cellophane film and transferred to a liquid MMN medium for another 30 days. Then, 50% PEG-6000 was added to the drought-stress treatment group (PEG) until the final concentration reached 10% PEG- 6000, and the mycelia were collected after 24 h of stress treatment. Meanwhile, 12.5 mL of liquid MMN medium without PEG-6000 was added to each flask of the control group. The mycelia were washed with 0.01 M PBS solution, frozen in liquid nitrogen, and stored at −80 °C before RNA isolation.

### 2.6. RNA-Seq Analysis

The mycelia of *C. geophilum* were grounded to powder in liquid nitrogen. Total RNA was extracted from 100 mg mycelia of each sample using the Trizol method. The quality of the RNA extract was determined by agarose gel electrophoresis.

A total of 1 µg of RNA per sample was used for the construction of a cDNA library. The construction of the cDNA library, quality verification, and further sequencing were performed by the Biomarker Technologies (Beijing, China) using an Illumina NovaSeq 6000 platform in accordance with standard protocols. The clean reads were mapped to the *C. geophilum* strain 1.58 reference genome v2.0 (Cenge3, https://mycocosm.jgi.doe.gov/Cenge3/Cenge3.home.html; accessed on 26 January 2021) using the HISAT2 software with default parameters [41], then the reads were assembled into transcripts to compare with reference genes using the String Tie software [42]. Differentially expressed genes (DEGs) of mycelia between the drought treatment and control groups were identified and annotated using the DESeq2 tool [43]. |log2Foldchange| ≥ 1.5 and *p*-value < 0.05 were used as the screening criteria for DEGs. The novel genes identified in the enrichment analysis were ruled out in further analysis. Further Kyoto Encyclopedia of Genes and Genomes (KEGG) and Gene Ontology Consortium (GO) analysis were used to annotate and conduct the DEGs with the KEGG database and PlantGSEA software, respectively [44,45].

### 2.7. Experimental Validation of Gene Expression Using qRT-PCR

In our experiments, the samples analyzed by RNA sequencing were also used for qRT-PCR analysis. The expression of seven unigenes identified by RNA-seq was validated using qRT-PCR. Gene-specific primers were designed based on the unigene sequences (Appendix A). The 18S rRNA genes were amplified as endogenous loading controls for testing the validity of the template preparation [46]. The tool Premier 5.0 (http://www.premierbiosoft.com/; accessed on 2 June 2022) was used to design the primers for the candidate genes. The expression of each gene was confirmed in at least three rounds of independent qRT-PCR reactions.

### 2.8. Statistical Analysis

Excel 2010 was used to calculate the mean and standard deviation. In order to test the significant differences of physiological parameters between the control and experimental groups of different drought-tolerant strains, Welch’s *t*-test was performed using SPSS 21.0. The least significant difference (LSD) method was used for multiple comparisons, and the significance criterion was defined as *p* < 0.05. All experimental data were recorded from three biological replicates.

## 3. Results

### 3.1. Mycelial Growth of Different C. geophilum Strains under Drought Treatment

The mycelial growth of the three drought-sensitive strains (Chcg01, Jacg40, and Jacg45) were significantly inhibited by drought stress (Figure 1). Compared to the control, the mycelial area decreased to 32.1%, 54.7%, and 33.4%, respectively (Appendix A). The drought treatment slightly affected the mycelial growth of the two tolerant strains (Jacg37 and Jacg205), and significantly promoted the growth of Jacg153, with the area increasing to 134%.

### 3.2. Antioxidant Enzyme Activities of Different C. geophilum Strains under Drought Treatment

The effects of drought treatment on the antioxidant enzymes of *C. geophilum* are shown in Figure 2. The drought treatment significantly enhanced the activities of SOD, POD, and CAT in the mycelia of Chcg01 and Jacg45 (drought-sensitive) and the activities of SOD and POD in the mycelia of Jacg37 (drought-tolerant), respectively. The activities of these antioxidant enzymes in the mycelia of the remaining strains were not affected by the drought treatment.

### 3.3. Na, K, P, and Ca Concentrations of Different C. geophilum Strains under Drought Treatment

In order to estimate how the drought treatment influenced the nutrient uptake in the mycelia of *C. geophilum*, we measured the contents of sodium (Na), potassium (K), calcium (Ca), and phosphorus (P) in *C. geophilum* mycelia (Figure 3). The drought treatment did not affect the Ca content in the mycelia of any strain. The contents of Na, K, and P in the mycelia of the drought-sensitive Chcg01 and Jacg45 strains and the drought-tolerant Jacg37 strain were significantly increased by the drought treatment, whereas the Na, K, P, and Ca contents in the mycelia of the drought-tolerant Jacg153 and Jacg205 strains significantly decreased or did not change after the drought treatment.

### 3.4. DEG Expression of Different C. geophilum Strains under Drought Treatment

Transcriptome analysis was performed on the six strains after being cultured for 24 h under drought stress and non-drought stress to screen the key genes for drought tolerance of *C. geophilum.* After filtration, we obtained a total of 188.66 Gb of clean data. More than 94.93% of bases in raw reads have a Q value ≥ 30 and more than 98.00% of bases in clean reads have a Q value ≥ 20. At the same time, from the statistics of the comparison results, the GC content of all samples is about 51%, and the GC content of each sample is similar (Appendix A). All test parameters of the sample meet the quality requirements and are suitable for further analysis.

Using the DESeq software, we set fold change (FC) ≥1.5 and *p*-value < 0.01 as the selection criteria for DEGs, and the analysis identified a total of 602, 950, 782, 301, 635, and 639 significant DEGs in Chcg01, Jacg40, Jacg45, Jacg153, and Jacg205, respectively (Figure 4). There were 2334 DEGs in the three drought-sensitive strains, of which 1824 DEGs were up-regulated and 510 were down-regulated (Figure 4). Chcg01, Jacg40, and Jacg45 (drought-sensitive strains) shared 53 DEGs (Figure 5a), and the number of specific DEGs was 425, 515, and 367, respectively. Among the three drought-resistant strains, there were a total of 1575 DEGs, of which 995 DEGs were up-regulated and 580 DEGs were down-regulated (Figure 4). Jacg37, Jacg153, and Jacg205 shared 39 DEGs (Figure 5b), and the number of specific DEGs was 162, 453, and 457, respectively. Under drought stress, the drought-sensitive strains had more DEGs than the drought-tolerant strains. The heat maps of shared DEGs show that 90 percent of the shared DEGs are up-regulated and only 10 percent are down-regulated (Figure 5c,d).

### 3.5. Gene Ontology (GO) Analysis of DEGs

A gene ontology (GO) term enrichment analysis and a manual inspection of the identified DEGs were performed to identify the biological processes (BP), molecular function (MF), and cellular compartment (CC) that are mostly affected in the *C. geophilum* strains under drought stress (Figure 6). For drought-responsive DEGs, the GO items “metabolic process”, “single-organism process”, “cellular process”, “localization”, and “biological regulation”, which belonged to the biological process category; the GO items “membrane”, “membrane part”, “cell”, ”cell part”, and “organelle”, which belonged to the cellular component category; and the GO items “catalytic activity”, “binding”, “transporter activity”, and “nucleic acid binding transcription factor activity”, which belonged to the molecular function category, were enriched in both drought-sensitive strains and drought-tolerant strains. Meanwhile, the DEGs in the three drought-tolerant strains were enriched in “supramolecular complex” and “molecular function regulator”.

### 3.6. KEGG Pathway Enrichment of DEGs

To study the biological behavior of the DEGs involved in drought stress response, we mapped the DEGs to the reference canonical pathways in the Kyoto Encyclopedia of Genes and Genomes (KEGG) to further identify the active metabolic pathways involved in drought resistance (Figure 7). The three drought-sensitive strains share the pathways of peroxisome (ko04146), nitrogen metabolism (ko00910), tyramine acid metabolism (ko00350), fatty acid metabolism (ko01212), α-linolenic acid metabolism (ko00592), and taurine and hypotaurine metabolism (ko00430). Furthermore, we found that Chcg01 and Jacg40 have DEGs enriched in the cysteine and methionine metabolism pathway (KO00270). There are also some DEGs from Jacg40 and Jacg45 enriched in galactose metabolism (ko00052). The DEG enrichment pathways of the drought-tolerant strains are different from those of the drought-sensitive strains. The three drought-tolerant strains share the pathways of ubiquinone and other terpenoid-quinone biosynthesis (ko00130) and sphingolipid metabolism (ko00600). In addition, both Jacg153 and Jacg205 have DEGs annotated to ABC transporters (KO02010). Meanwhile, the DEGs of Jacg37 and Jacg205 are enriched in glutathione metabolism (KO00480).

### 3.7. Analysis of Key Genes in C. geophilum Resistance to Drought Stress

We created a heatmap of all DEGs enriched in the same pathway to select the key genes of *C. geophilum* resistance to drought stress (Figure 8). In the drought-sensitive strains, there are 20 DEGs enriched in the common pathway of peroxisome (ko04146), 9 DEGs enriched in nitrogen metabolism (ko00910), 4 DEGs enriched in α-linolenic acid metabolism (ko00592), 9 DEGs enriched in fatty acid metabolism (ko01212), 3 DEGs enriched in taurine and hypotaurine metabolism (ko00430), and 7 DEGs enriched in tyramine acid metabolism(ko00350) (Figure 8a). The K441DRAFT_186058, which is annotated into the peroxisome pathway of thee drought-sensitive strains, is all up-regulated after the drought treatment. The K441DRAFT_702334 is significantly up-regulated in the fatty acid metabolism, α-linolenic acid metabolism, and peroxisome pathways in the drought-sensitive strains (Figure 8a). There are six DEGs enriched in ubiquinone and other terpenoid-quinone biosynthesis (ko00130) and three DEGs enriched in sphingolipid metabolism (ko00600) in the drought-tolerant strains (Figure 8b). Two genes in the ubiquinone and other terpenoid-quinone biosynthesis pathways are significantly enriched in the three drought-tolerant strains, including K441DRAFT_685815 and K441DRAFT_660991, which are annotated as 4-coumarate-CoA ligase-like protein (4CL) and benzoquinone reductase (Figure 8b).

### 3.8. RT-qPCR Verification of DEGs

In order to verify the reliability of the sequencing results, we selected two genes in the ubiquinone and other terpenoid-quinone biosynthesis pathway (K441DRAFT_685815; K441DRAFT_660991), three genes in the peroxisome pathway (K441DRAFT_702334; K441DRAFT_186058; K441DRAFT_614568), one gene in the lipid transport and metabolism pathway (K441DRAFT_656193), and a significantly down-regulated gene (K441DRAFT_571428) for real-time quantification polymerase chain reaction (RT-qPCR) validation. The relative expression levels of these genes are presented in Figure 9. The results show that K441DRAFT_685815 and K441DRAFT_660991 are significantly upregulated in the drought-tolerant strains Jacg37 and Jacg153. In the drought-sensitive strains, K441DRAFT_702334, K441DRAFT_186058, and K441DRAFT_614568 are significantly up-regulated after the drought treatment. Moreover, the gene K441DRAFT_571428 is significantly down-regulated after the drought treatment. These results are consistent with the RNA transcriptome sequencing, indicating that the RNA-Seq results are valid for analyzing the mechanism of *C. geophilum* response to drought stress.

## 4. Discussion

*Cenococcum geophilum* is one of the most frequently encountered ectomycorrhizal fungi in nature [32]. Worley and Coleman et al. showed that ectomycorrhizal fungus *C. geophilum* is highly drought tolerant [47,48]. However, the mechanisms and genes involved in the drought-tolerant regulation of *C. geophilum* remain unclear. Our results showed that the drought-sensitive strains responded to drought stress by increasing the contents of Na, K, and P. At the same time, the genes related to peroxisome synthesis were up-regulated and the content of antioxidant enzymes increased in the drought-sensitive strains. However, the drought-tolerant strains resisted drought stress by up-regulating the genes involved in the ubiquinone and other terpenoid-quinone biosynthesis pathway. Moreover, the drought-tolerant strains up-regulated the key fulcrum enzyme 4-coumarate-CoA ligase (4CL) in the phenylpropane pathway in response to drought stress.

Previous studies have shown that the accumulation of Na can significantly reduce osmotic potential and enhance the uptake of water in the xerophyte *Zygophyllum xanthoxylum* during drought treatment [49,50]. Potassium (K) is an essential nutrient element in plants, and plays a key role in many fundamental processes, thus affecting the growth and development of the whole plant [51,52,53]. In addition, more studies have proved that the accumulation of K can enhance the osmotic adjustment (OA) to promote the water acquisition of plants. Phosphorus (P) is a macronutrient, so its deficiency limits plant growth [54,55]. Hammer et al. showed that the contents of Na, K, Ca, and P in mycorrhizal fungi usually changed after salt stress [56]. In our results, the contents of Na, K, and P increased significantly in the drought-sensitive strains of Chcg01 and Jacg45 after drought stress. Therefore, we speculate that the drought-sensitive strains regulate the osmotic potential of mycelium and promote water absorption by accumulating Na and K. The change trend of nutrient uptake of the drought tolerant strain Jacg37 was consistent with those of the sensitive strains. We guess that the mycelia of Jacg37 might also suffer the drought stress in the treatment of 10% PEG, although the mycelial growth area of Jacg37 did not change. The K and P contents of mycelia of the drought-tolerant Jacg205 was significantly decreased by the drought treatment. Previous studies in plants suggest that a decrease in the concentration of Na, K, and P results from relatively slow growth, smaller absorption area of roots [57], and slow ion flow [58]. In the present study, different from plants, the drought treatment markedly enhanced the mycelial growth of Jacg205 (134%). Therefore, the concentration decrease of K and P in the mycelia of Jacg205 seems to be the dilution effect of mycelial growth. In addition, the drought treatment did not affect the Ca content of the six strains. Li et al. [35] also reported that the Ca contents in the mycelia of *C. geophilum* did not change under the salt stress. These results indicate that the Ca transport of mycelia of *C. geophilum* may hardly be affected by different stresses.

Peroxisomes play vital roles in metabolism since they are involved in many processes, including fatty acid β-oxidation, glyoxylate cycles, and biosynthesis of phytohormones, such as indole-3-acetic acid, jasmonic acid, and salicylic acid [59,60,61]. In addition, peroxisomes are able to produce antioxidant enzymes and reactive oxygen species to maintain intracellular redox homeostasis [18]. Studies have found that abiotic stresses (drought and salt) can induce an increase in peroxisome content in plants [62,63,64]. Our results showed that the activity of antioxidant enzymes (CAT, SOD, and POD) of the drought-sensitive strains increased significantly under drought stress. These results indicated that the increase in antioxidant enzymes was the way of the drought-sensitive strains’ response to drought. In addition, DEGs were enriched in the peroxisome pathway and significantly up-regulated in the drought-sensitive strains after drought stress (Figure 7). In particular, the peroxisome membrane protein gene (PMP34) and peroxin protein-encoded genes (PEXs) were significantly up-regulated. It has been reported that peroxin proteins encoded by PEX genes are required for peroxisome biogenesis [60,65], and peroxisome membrane proteins (PMPs) are the key to generate peroxisomes [61]. These results suggest that an up-regulation of key genes of peroxisome synthesis and enhancement of antioxidant enzyme activity are the main response mechanisms of the drought-sensitive strains to drought stress. Therefore, we speculate that, under drought stress, *C. geophilum* can up-regulate the gene encoding peroxisome synthesis to produce peroxisomes, and then secrete antioxidant enzymes to alleviate the damage of intracellular reactive oxygen species.

Sphingolipids are composed of three parts: long-chain (sphingoid) base (LCB), headgroup, and acyl chain [66]. As the main components of plasma membrane, sphingolipids can cluster with sterols in the plasma membrane to form lipid rafts, which are enriched in proteins, to alleviate abiotic stress [67,68]. The formation of lipid rafts depends on the modifications occurring in the sphingolipid structure, such as fatty acid chain length, fatty acid, and LCB hydroxylation [69]. Zhang et al. overexpressed long-chain basic kinase (OsLCBK1) in tobacco, which improved the tolerance of tobacco to oxidative stress and induced the expression of oxidative stress genes [70]. The LCB can be linked with long-chain fatty acids through amide bonds to form ceramides. Li et al. showed that *Arabidopsis* overexpressing neutral ceramidase (nCer1) were more tolerant to oxidative stress [71]. In this experiment, the K441DRAFT_592538 encoding ceramidase and the K441DRAFT_554224 encoding sphingosine hydroxylase were significantly up-regulated after drought stress in the drought-tolerant strains. Therefore, we conclude that Sphingolipid metabolism is an important pathway for *C. geophilum* to alleviate drought stress.

Ubiquinone (UQ), as an antioxidant, plays an indispensable role in plant growth and development by participating in the biosynthesis, metabolism of important compounds, and response of plants to stress [72]. Moreover, UQ can prevents DNA damage and cell membrane lipid peroxidation through the elimination of ROS [72]. Ohara et al. found that transgenic plants with higher UQ levels had a strong tolerance to oxidative damage caused by high salinity stress [73]. Chang et al. showed that the redox transition of UQ played a key role in cell regulation under low-temperature stress [74]. In addition, UQ can regulate the adaptability of Arabidopsis to high oxidative stress environment [75]. Compared to the drought-sensitive strains, ubiquinone and other terpenoid-quinone biosynthesis was enriched in the drought-tolerant strains (Figure 7), suggesting that *C. geophilum* is able to regulates genes involved in UQ synthesis in response to drought stress.

Phenylpropane metabolism is one of the most important secondary metabolic pathways in plants, which plays an important role in plant development and plant–environment interaction [76]. 4-coumarate-CoA ligase (4CL) is a key enzyme in the UQ synthesis pathway and an important fulcrum enzyme in the phenylpropane pathway, which can regulate plant growth and resist biotic and abiotic stresses [77,78,79]. For example, the overexpression of 4CL in tobacco improved its drought resistance due to increased lignin accumulation and antioxidant enzyme activity, and up-regulated the expression levels of stress-related genes [80]. Moreover, the overexpression of 4CL could improve the tolerance of transgenic Arabidopsis to drought stress [81]. In our results, the 4CL gene was significantly up-regulated after drought stress in the drought-tolerant strains, which is consistent with the findings of others, indicating that the up-regulation of 4Cl is the response of the drought-tolerant strains to drought stress.

## 5. Conclusions

This study explored the physiological and transcriptional responses of *C. geophilum* to drought stress stimulated by 10% polyethylene glycol (PEG-6000). Our results showed that the drought-sensitive strains of *C. geophilum* might absorb Na and K ions to regulate osmotic pressure, and up-regulate peroxisome pathway genes to promote the activity of antioxidant enzymes to alleviate drought stress, while the drought-tolerant strains resist drought stress by up-regulating the functional genes involved in the ubiquinone and other terpenoid-quinone biosynthesis and sphingolipid metabolism pathways. In addition, the up-regulation of 4CL could enhance the metabolites in the phenylpropane pathway and improve the tolerance of drought-tolerant *C. geophilum* to drought stress. The results provide new perspectives into the molecular mechanism of *C. geophilum* response to drought treatment, which is helpful to study the drought stress response of ectomycorrhizal fungi.

## Figures and Tables

**Figure 1 jof-09-00015-f001:**
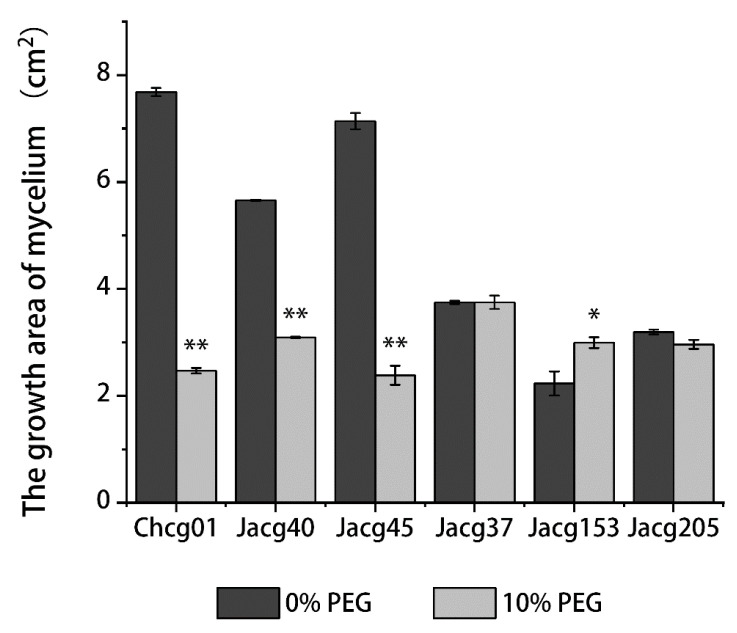
The mycelial growth of six strains of *Cenococcum geophilum* under drought treatments. The mycelia were pre-cultured in a modified MMN medium for 30 days, followed by the drought treatment (10% PEG-6000) for 30 days. PEG-6000 was not added into the control treatment. Data and bars are shown as mean and ± SD of the replicates, respectively (n = 3). Significant difference is analyzed by using the Student’s *t*-test. * *p* < 0.05; ** *p* < 0.01.

**Figure 2 jof-09-00015-f002:**
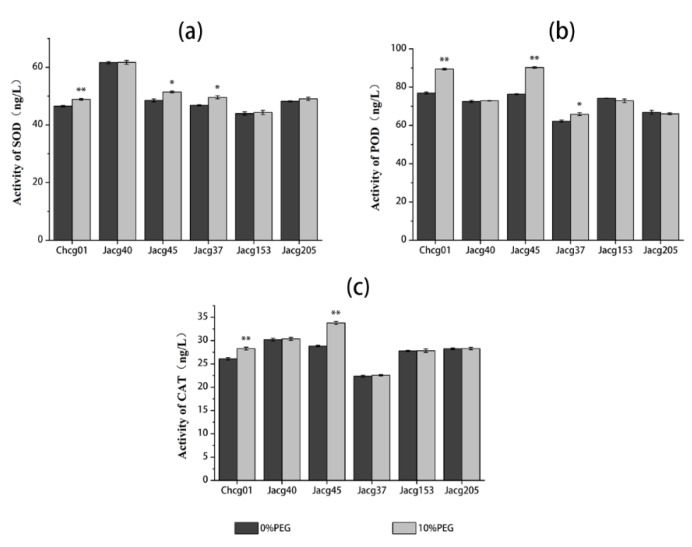
The activities of superoxide dismutase (SOD, (**a**)), peroxidase (POD, (**b**)), and catalase (CAT, (**c**)) in the mycelia of six strains of *Cenococcum geophilum* under drought treatment. The mycelia were pre-cultured in a modified MMN medium for 30 days, followed by the drought treatment (10% PEG-6000) for 30 days. PEG-6000 was not added into the control treatment. Data are shown as mean ± SD of the replicates (n = 3). Significant difference of the activities of SOD, POD, and CAT after the drought treatment is analyzed by using the Student’s *t*-test. * *p* < 0.05; ** *p* < 0.01.

**Figure 3 jof-09-00015-f003:**
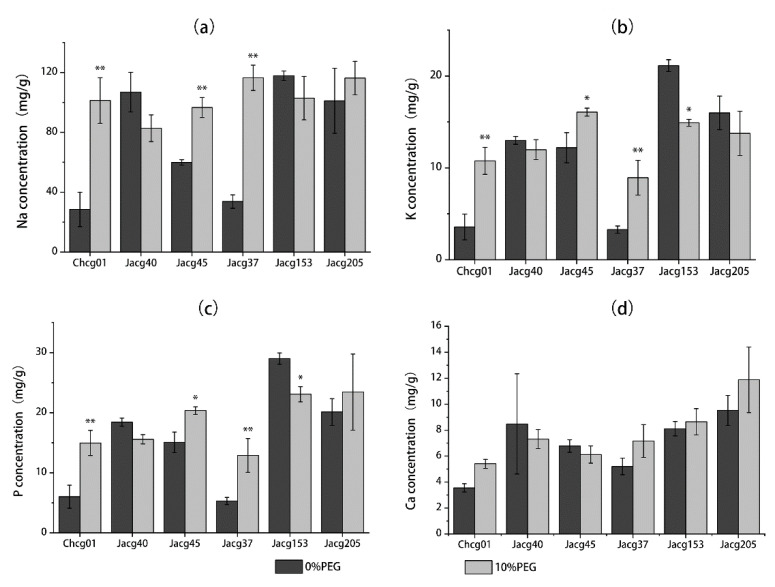
The sodium (Na, (**a**)), potassium (K, (**b**)), phosphorus (P, (**c**)) and calcium (Ca, (**d**)) concentrations in the mycelia of six strains of *Cenococcum geophilum* under drought treatment. The mycelia were pre-cultured in a modified MMN medium for 30 days, followed by the drought treatment (10% PEG-6000) for 30 days. PEG-6000 was not added into the control treatment. Data are shown as mean ± SD of the replicates (n = 3). Significant difference of each element concentration of each isolate between 0% and 10% PEG-6000 treatments is tested by using the Welch’s *t*-test. * *p* < 0.05; ** *p* < 0.01.

**Figure 4 jof-09-00015-f004:**
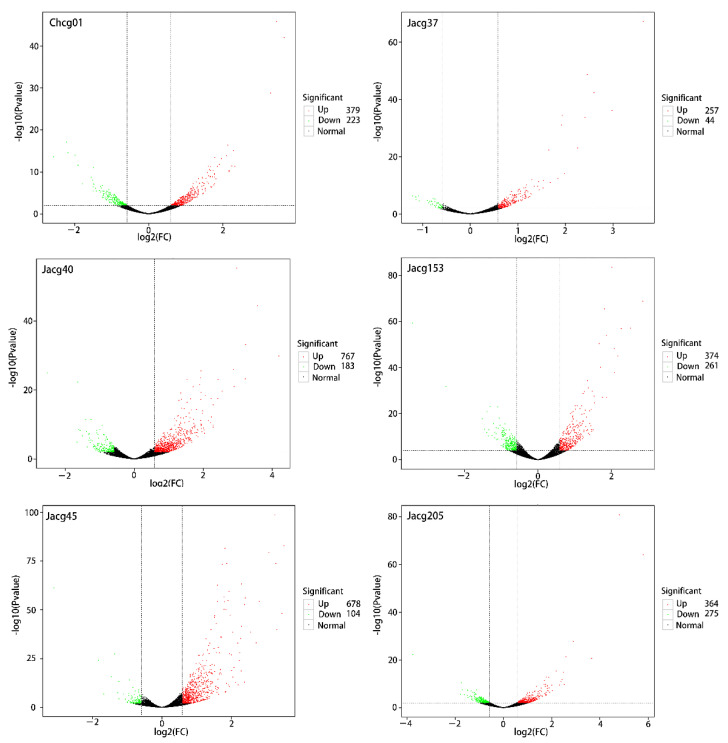
Volcano map of the candidate genes induced by the drought treatments. Volcano map shows the differentially expressed genes (DEGs) of six *Cenococcum geophilum* after 10% PEG-6000 treatment compared to 0% PEG-6000. Red spot indicates significant up-regulated genes after drought treatment; green spot indicates significantly down-regulated genes; and black spot indicates no-change genes. |log2Foldchange| ≥ 1.5 are used as the screening criteria of DEGs.

**Figure 5 jof-09-00015-f005:**
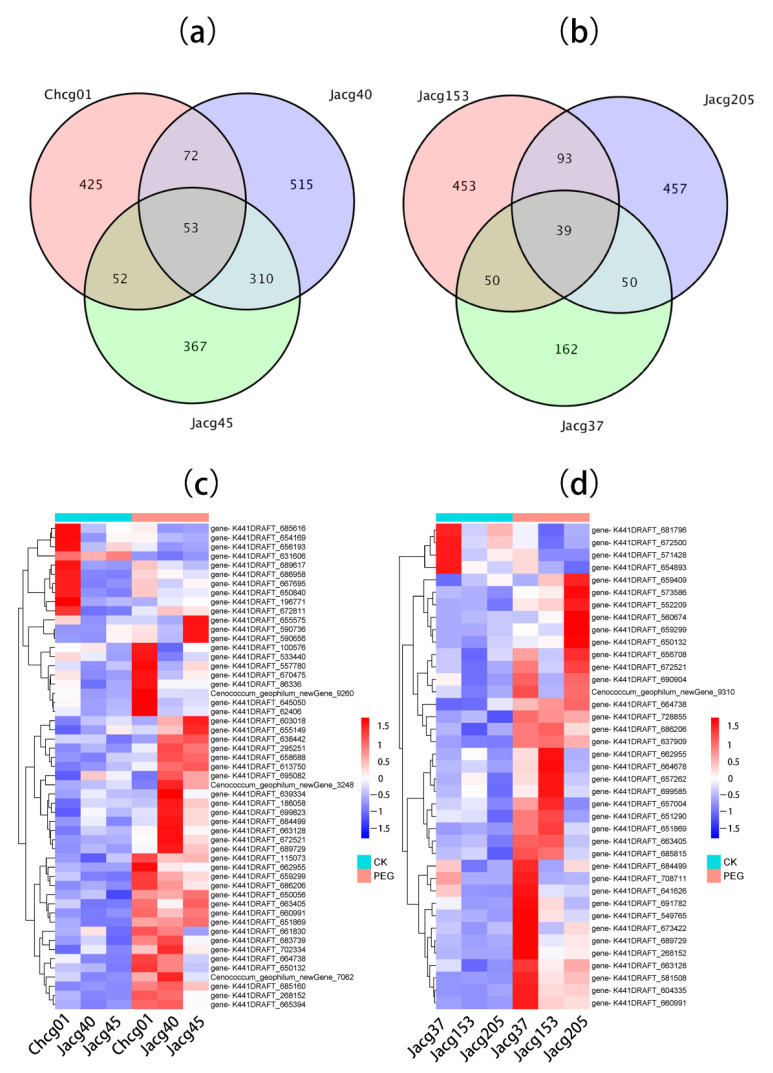
Genome analysis of the DEGs in six strains of *Cenococcum geophilum* between the drought (10% PEG 6000, PEG) and the control treatments (0% PEG 6000, CK). (**a**,**b**) indicate the Venn diagrams of total DEGs in the 3 drought-sensitive strains (Chcg01, Jacg40, and Jacg45) and the 3 drought-tolerant strains (Jacg37, Jacg153, and Jacg205) of *Cenococcum geophilum* after 10% PEG-6000 treatment compared to the control treatment (0% PEG 6000), respectively. (**c**,**d**) indicate the heatmaps of 53 common genes for the Chcg01, Jacg40, and J45 strains and 14 common genes for the Jacg37, Jacg153, and Jacg205 strains after drought treatment, respectively. The different colors of the heatmap, ranging from blue over white to red, represent scaled expression levels of genes with [log2 (FPKM + 1)] across different samples.

**Figure 6 jof-09-00015-f006:**
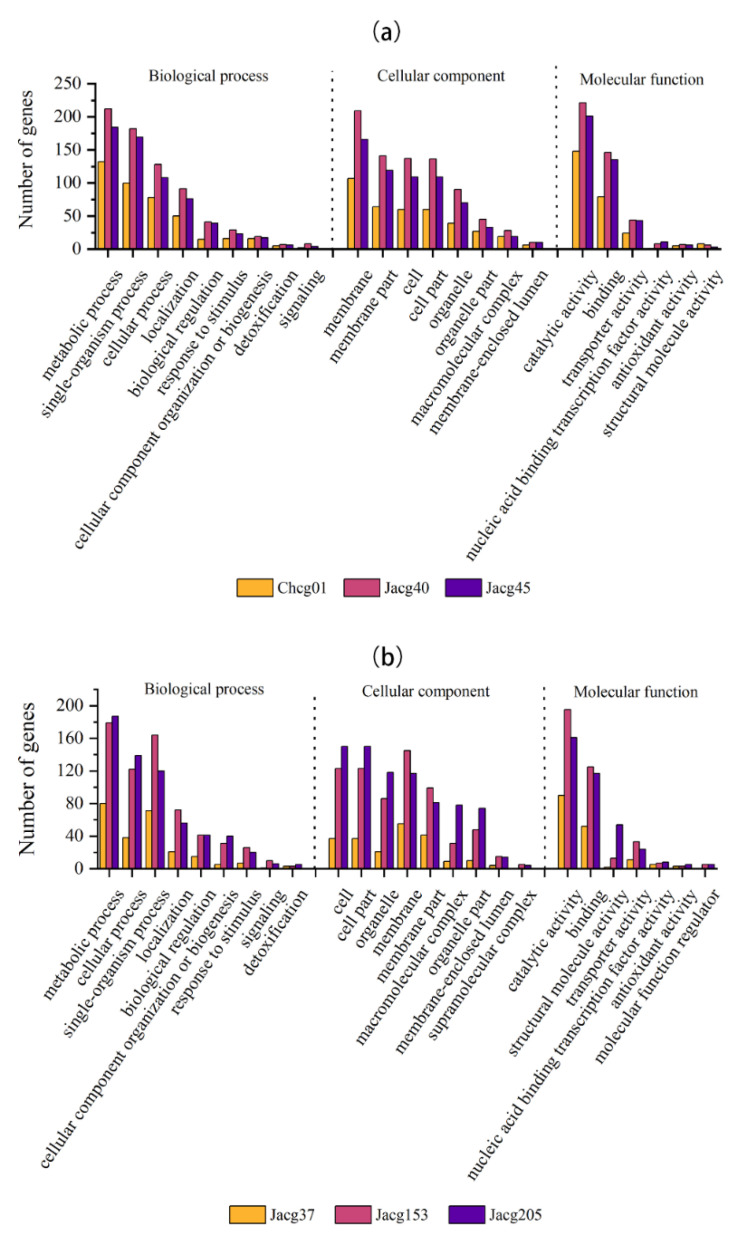
Gene ontology (GO) enrichment analysis of the DEGs in six strains of *Cenococcum geophilum* after 10% PEG-6000 treatment compared to the control treatment (0% PEG-6000). (**a**) three drought-sensitive strains (Chcg01, Jacg40, and Jacg45); (**b**) three drought-tolerant strains (Jacg37, Jacg153, and Jacg205). The abscissa is the GO classification and the ordinate is the number of genes.

**Figure 7 jof-09-00015-f007:**
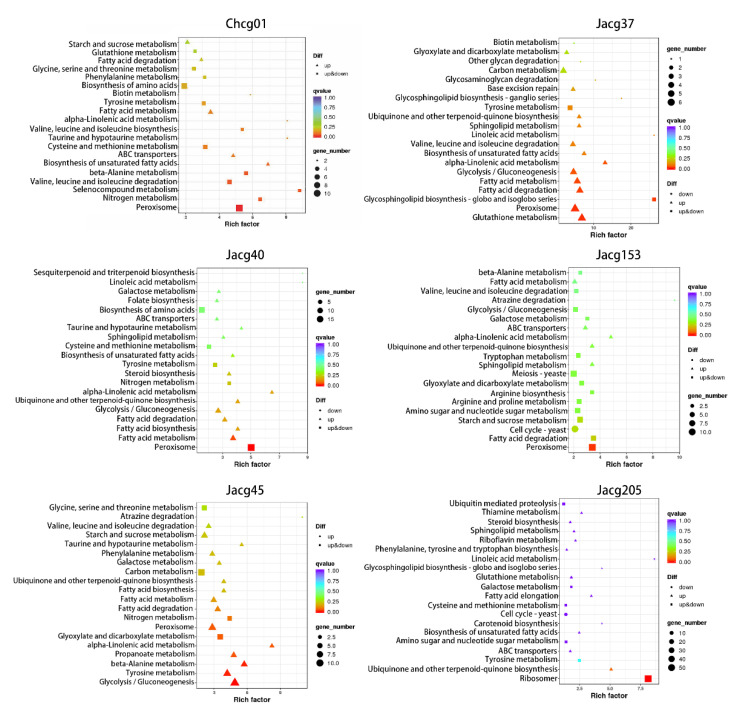
Kyoto Encyclopedia of Genes and Genomes (KEGG) enrichment analysis of the DEGs in six strains (Chcg01, Jacg37, Jacg40, Jacg45, Jacg153, and Jacg205) of *Cenococcum geophilum* after 10% PEG-6000 treatment compared to the control (0% PEG-6000). The size of the symbol represents the numbers of DEGs involved in the corresponding pathway.

**Figure 8 jof-09-00015-f008:**
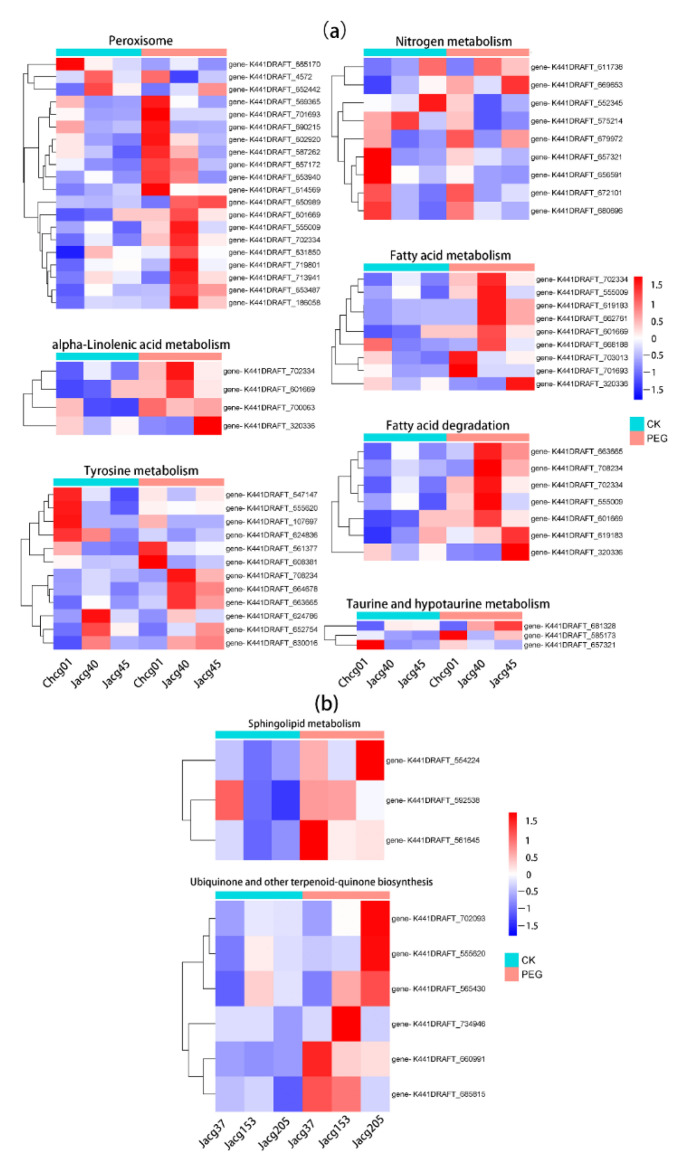
The heatmap of DEGs in the different pathways in six strains of *Cenococcum geophilum* after 0% and 10% PEG-6000 treatments. (**a**), sensitive strains; (**b**), tolerant strains. The Y- and X-axes represent the differentially expressed genes and different samples, respectively. The different colors of the heatmap, ranging from blue over white to red, represent scaled expression levels of genes with [log2 (FPKM + 1)] across different samples.

**Figure 9 jof-09-00015-f009:**
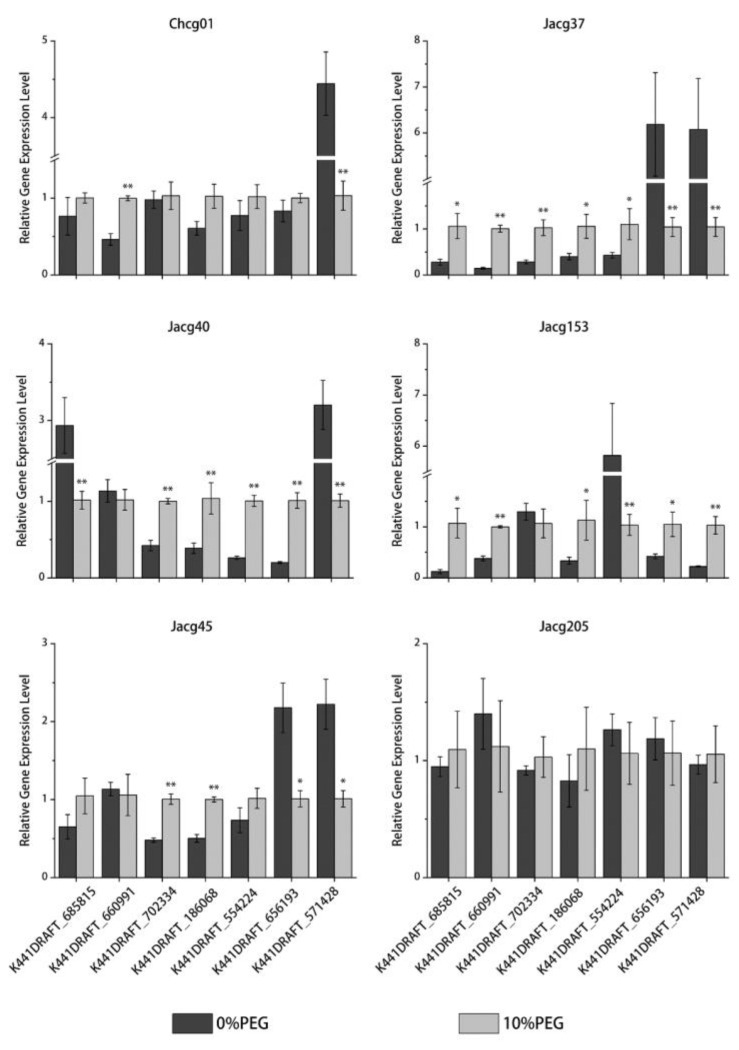
The relative expression levels of seven candidate genes in six strains of *Cenococcus geophilum* analyzed using qRT-PCR after 0% and 10% PEG-6000 treatments. *Cenococcum geophilum* 18S gene was used as the internal control, and 2^−ΔΔCt^ method was used to evaluate gene expression profile. Three biological replicates and three technical replicates were performed for each sample. The statistically significant difference between 0% and 10% PEG-6000 treatments is tested by Student’s *t*-test (* *p* < 0.05; ** *p* < 0.01).

## Data Availability

The data presented in this study are available within the article and Appendix A. Data for *Cenococcum geophilum* raw sequence reads are available in a publicly available repository [NCBI], reference number [PRJNA884483].

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
