# Peer review of "The Transcriptional Responses of Ectomycorrhizal Fungus, Cenococcum geophilum, to Drought Stress"

_jof, 2022, doi:10.3390/jof9010015_

Round 1
Reviewer 1 Report (Previous Reviewer 3)
The suggestions in my review were taken into account, although I still have some concerns about replication in the experimental design and about the writing style.
Author Response
Dear Reviewer:
Thank you for your professional review work on our manuscript (Manuscript ID: jof-2077976). All your suggestions were very valuable for improving our paper. We have revised the manuscript accordingly. A point-by-point response is provided in the following. All changes are marked with red color in the revised manuscript.
Comment 1: The suggestions in my review were taken into account, although I still have some concerns about replication in the experimental design and about the writing style.
Response 1: Thank you for your comments. Three replicates were analysed for the determination of each physiological index in this experiment.

Reviewer 2 Report (New Reviewer)
Minor comments
1)Line 129- use a new paragraph while speaking about Na,Ca,P,K determination
2)Line 133- Write PerkinElmer using capical letter
3)Line 246 ‘under cadmium treatment’- is it a misprint?
4)Line 277 ‘the horizontal coordinate’- may be abscissa? And ‘vertical coordinate is ‘ordinate’. Please, make changes
5)Discussion: Participation of sphingolipids in drought tolerance of ectomycorrhizal fungi is extremely valuable. Are you able to provide a small discussion on the topic?
6)Reference list- please, use journals’ abbreviations
7)Refs 35,36,37- use italics for ‘Cenococcum geophilum’, and below check all the references
Author Response
Dear Reviewer:
Thank you for your professional review work on our manuscript (Manuscript ID: jof-2077976). All your suggestions were very valuable for improving our paper. We have revised the manuscript accordingly. A point-by-point response is provided in the following. All changes are marked with red color in the revised manuscript.
Review 2:
Point 1: Line 129- use a new paragraph while speaking about Na,Ca,P,K determination
Response: Thank you for your suggestion. We follow your suggestion with a new paragraph describing the method of Na, Ca, P and K determination. Please check in L132-138.
Point 2: Line 133- Write PerkinElmer using capical letter
Response: Thank you for your comments. I am very sorry for the mistake and I have corrected in the L135.
Point 3: Line 246 ‘under cadmium treatment’- is it a misprint?
Response: Thank you for your reminding. We've changed it to “drought management”. Please check in L247.
Point 4: Line 277 ‘the horizontal coordinate’- may be abscissa? And ‘vertical coordinate is ‘ordinate’. Please, make changes
Response: Thank you for your comments. We have changed it to “abscissa and ordinate” in the manuscript according to your suggestion. Please check in L281.
Point 5: Discussion: Participation of sphingolipids in drought tolerance of ectomycorrhizal fungi is extremely valuable. Are you able to provide a small discussion on the topic?
Response: Thank you a lot for your comments. We have added a discussion on "sphingolipid metabolism" to the discussion section of the manuscript.
Sphingolipids are composed of three parts: long chain (sphingoid) base (LCB), headgroup and acyl chain. As the main components of plasma membrane, sphingolipids can cluster with sterols in the plasma membrane to form lipid rafts, which were enriched in proteins, to alleviate abiotic stress. The formation of lipid rafts depends on the modifications occurring in the sphingolipid structure, such as fatty ac-id chain length, fatty acid and LCB hydroxylation. Zhang et al. overexpressed long chain basic kinase (OsLCBK1) in tobacco, which improved the tolerance of tobacco to oxidative stress and induced the expression of oxidative stress genes. The LCB can be linked with long-chain fatty acids through amide bonds to form ceramides. Li et al. showed that Arabidopsis overexpressing neutral ceramidase (nCer1) were more tolerant to oxidative stress. In this experiment, the K441DRAFT_592538 encoding ceramidase and the K441DRAFT_554224 encoding sphingosine hydroxylase were significantly upregulated after drought stress in the drought-tolerant strains. Therefore, we conclude that Sphingolipid metabolism is an important pathway for C. geophilum to alleviate drought stress. Please check in L406-419.
Point 6: Reference list- please, use journals’ abbreviations
Response: Thank you for your comments. We have changed the names of all journals in the references to abbreviations.
Point 7: Refs 35,36,37- use italics for ‘Cenococcum geophilum’, and below check all the references
Response: Thank you for your reminding. I've checked all the “Cenococcum geophilum” in the manuscript and set them in italics.

This manuscript is a resubmission of an earlier submission. The following is a list of the peer review reports and author responses from that submission.
Round 1
Reviewer 1 Report
The manuscript describes an interesting work on the impact of drought (probably in this specific case it should be better define water stress or water limitation) on the mycelium of an ECM fungus by transcriptomics. I appreciate the performed work considering that it was proposed that Cenococcum could influece plant response to drought changing aquaporins' expression. I suggest the authors to carefully check the paper of the 2016 by Peter et al. on Nature Communications. About the M&M please add all the considered replicates in this section (not only in the paragraph on statistics), rendering immediately clear the robustness of the performed work. Figure 6, related to GO, has the words too small. Please try to improve this point since in this way it is very difficult to follow the results. About RT-qPCR, in my opinion Table S3 should be imporved adding the names of the considered genes (and not only the codes). It is important also to know if primers derived or not from previous published papers. As a major concern, I am not sure that the use of all the samples used in the sequencing without any novel samples (at least one) could be considered a good validation. Could the authors perform an additional set of RT-qPCR? It is also important that at least two housekeeping genes would be used.
Reviewer 2 Report
Li et al. Cenococcum
Mycelium growth
The mycelium growth of the six strains under 10% PEG-6000) for 30 days clearly differs between the resistant and the non-resistant strains to drought.
Figure 1 Please correct the legend
Relative mycelial growth rate of six strains of Cenococcum geophilum under 10% PEG-6000 for 30 days. (There is only one treatment and not several).
Antioxidant enzyme activities and Na, Ca, P and K
Results Figure 2 No clear differences of antioxidant activities between the resistant and the non-resistant strains
Na, Ca, P and K
Same as: No clear differences in Na, Ca, P and K uptake between the resistant and the non-resistant strains.
Figure 3, Legend: One treatment not different drought treatments.
Gene expression
Figure 4: indicate what are the resistant and not resistant strains. Difficult to see a difference of behavior between the two types of stains.
Figure 5: same conclusion: difficult to see significant differences.
Figure 6: too complicated and difficult to understand.
Figure 8: difficult to draw a conclusion.
Discussion
This discussion is confusing and does not adequately account for the lack of differences between strains for most of the factors analyzed.
Conclusion
The conclusion does not really reflect the results. It is extremely difficult to draw clear conclusions and to really differentiate the differences in behavior between resistant and non-resistant strains. This is a result, admittedly negative, but interesting.
Reviewer 3 Report
The authors analyze the response of Cenococcum geophilum to droughts in several aspects: production of enzymes, nutritional state and transcriptional response. This ectomycorrhizal fungus is widespread and particularly abundant in stressful environmental conditions. The research is timely and relevant, since global warming has increased the impact of droughts in many regions of the world. The authors make an important effort to evaluate the existence of drought-tolerant strains of C. geophilum and to characterize the transcriptional mechanisms linked to these strains.
In general, the introduction, the objectives, the methodologies, the results and the discussion of the study are clearly explained and sufficiently detailed. However, the manuscript is hindered by several minor issues:
- Figs 2-3 seem to be based not in real replicates (the strain cultivated in different samples) but pseudo-replicates, as hinted in L133 (the same sample was measured 3 times). If this is so, Figs 2-3 should not have confidence intervals and tests of significant differences, because there is only one sample per strain. In any case, if the authors would like to calculate statistics, they could put together the data of the 3 tolerant strains and compare them with the pooled data of the 3 sensitive strains.
- Some figures of the supplementary information are not cited, and others are not in the order in which they are cited in the manuscript
- Sometimes hyphenation seems to be incorrectly used (e.g. L54, L61, etc.)
- Sometimes verbal forms are incorrect (e.g. L41 attracted, L72 scavenging, etc)
- In the results section it is frequent to find sentences such as "The ... were shown in Figure 1", when this could be avoided. E. g. "The mycelial growth of the sensitive strains was significantly inhibited by drought stress (Fig. 1, Supplemental Table S1)" in L182-184 would allow to eliminate the previous sentence (L181-182).
- L196 reference to Figure 3 seems to be incorrect. Should it be Figure 2?
- Could it be possible to improve the readibility of Figure 6?
- L331 reference to Figure 8 seems to be incorrect. Should it be Fig. 9?
- L430 "The drought-tolerant strains might up-regulate...". In the conclusions there should be no discussion but results of the study. Maybe better to say that "The analyzed drought-tolerant strains up-regulate..."